# Extraction of Naringin from Pomelo and Its Therapeutic Potentials against Hyperlipidemia

**DOI:** 10.3390/molecules27249033

**Published:** 2022-12-18

**Authors:** Xiaolei Yu, Xin Meng, Yidi Yan, Hui Wang, Lei Zhang

**Affiliations:** 1MOE Key Laboratory for Nonequilibrium Synthesis and Modulation of Condensed Matter, School of Physics, Xi’an Jiaotong University, Xi’an 710049, China; 2Meat Processing and Safety Control Professional Technology Innovation Center, Jinzhou Medical University, Jinzhou 121000, China

**Keywords:** anti-hyperlipidemic, organ index, naringin, pomelo peel, process condition

## Abstract

Pomelo peel is a natural plant product with numerous pharmacological effects and is used in traditional Chinese medicine. In the present study, we extracted naringin from pomelo peel and aimed to decipher its therapeutic potential against hyperlipidemia. We used ultrasonic-assisted extraction to obtain naringin prior to identifying its structure, to evaluate its ability in binding sodium glycine cholate and sodium bovine cholate in vitro by simulating the gastrointestinal environment, so as to evaluate its blood lipid-lowering activity. The hyperlipidemia mouse model was established. Following the intragastric administration of naringin for 5 weeks, we measured the weight change, organ index, high-density lipoprotein cholesterol (HDL-C), serum total cholesterol (TC), serum triglycerides (TG), liver superoxide dismutase (SOD), glutathione peroxidase (GSH-Px), low-density lipoprotein cholesterol (LDL-C) level, malondialdehyde (MDA), alanine aminotransferase (ALT), and aspartate aminotransferase (AST) level of mice in the normal control and high-fat diet groups in addition to the high-, medium-, and low-dose naringin groups. The pathological changes in the liver were observed under a light microscope. The total RNA of the liver was extracted, and the mRNA expression level of lipid metabolism-related factors in mouse liver was detected via a fluorescence quantitative polymerase chain reaction (PCR). Naringin significantly (*p* < 0.01) reduced the body weight, organ index, serum TG, LDL-C, and TC levels of hyperlipidemic mice, but increased the serum HDL-C levels (*p* < 0.01). Furthermore, naringin increased GSH Px and SOD activity (*p* < 0.01), while decreasing MDA, ALT, and AST levels, as well as the liver index (*p* < 0.01). There was no statistically significant difference in the brain, heart, spleen, kidney, and other indicators (*p* > 0.05). A histopathological analysis of mouse liver showed that naringin could alleviate the degenerative damage of fatty liver cells in hyperlipidemic mice. Naringin could significantly (*p* < 0.01) reduce the expression of FAS and SREBP-1c mRNA, and simultaneously increase PPARα mRNA expression. This study shows that naringin has the strong effect of lowering lipids and protecting the liver in hyperlipidemic mice. Our findings underscore the anti-hyperlipidemia potential of naringin and increase the scientific understanding of its anti-hyperlipidemia effects, that may lead to its potential application as a dietary strategy for hyperlipidemia management in the future.

## 1. Introduction

Pomelo (*Citrus grandis* L.) is a citrus plant of the Rutaceae family, cultivated in China for over 3000 years. It is grown in Zhejiang, Jiangxi, Guangdong, Guangxi, Taiwan, Fujian, Hunan, Hubei, Sichuan, Guizhou, Yunnan, and other provinces. The pomelo prefers warm, humid climates and does not tolerate drought. The optimum temperature range for its growth is 23–29 °C, but it can nonetheless endure temperatures as low as −7 °C. Its fruits, flowers, and leaves contain a high-quality aromatic oil. The fruit pulp, which is processed into juice, wine, citric acid, and jam, contains high levels of vitamin C and favors digestion and detoxification. Pomelo peel has been prescribed in China for centuries for its anti-tussive, expectorant, anti-inflammatory, and other pharmacological effects. Pomelo peel is the immature or nearly mature dried exocarp of *Citrus grandis* “Tomentosa” or *Citrus grandis* (L.) Osbeck. The peel of the immature summer pomelo is typically cut into pieces, pressed, and dried after part of the mesocarp is removed. Pomelo peel may be soaked or decocted with water. Ethanol extracts of pomelo peel are used in certain TCM prescriptions [1]. Recent studies established that naringin is the main flavonoid constituent in pomelo peel. Naringin has anti-tumor, anti-allergy, anti-bacterial, anti-inflammatory, anti-microbial, and anti-mutagenic properties; protects damaged myocardium; and promotes thyroid cancer cell apoptosis, among other effects [2,3].

Hyperlipidemia is a metabolic disease characterized by abnormal lipid metabolism; elevated serum low-density lipoprotein cholesterol (LDL-C), triglyceride (TG), and total cholesterol (TC) levels; and reduced high-density lipoprotein cholesterol (HDL-C) levels [4,5]. Hyperlipidemia is a risk factor that promotes various cardiovascular and cerebrovascular diseases including atherosclerosis. At the beginning of the 20th century, cardiovascular disease was associated with less than 10% of the total global mortality rate. However, at the beginning of the 21st century, the mortality rate of cardiovascular disease has accounted for nearly 50% of the mortality rate in developed countries, and about 25% in developing countries [6,7]. It is a leading cause of metabolic diseases such as diabetes, hypertension, and fatty liver disease. Therefore, its prevention and treatment have an important therapeutic significance [8,9]. In this study, we purified naringin from the pomelo peel and evaluated its hypolipidemic properties and anti-oxidant capacity. Our results demonstrate the importance of the development and utilization of distinctive medical foods with anti-oxidant and lipid-lowering effects, specifically, pomelo peel resources.

## 2. Materials and Methods

### 2.1. Reagents and Raw Materials

Naringin (purity > 98%) was purchased from Hefei Bomei Biotechnology Co. Ltd. (Hefei, China). D101 macroporous resin was acquired from Solebo Biotechnology Co. Ltd. (Beijing, China). Test kits to measure total cholesterol (Cat. No. A111-2-1), triglyceride (Cat. No. A110-1-1), malondialdehyde (Cat. No. A003-1-1), glutathione peroxidase (Cat. No. A005-1-1), superoxide dismutase (Cat. No. A001-1-1), low-density lipoprotein cholesterol (Cat. No. A113-1-1), high-density lipoprotein cholesterol (Cat. No. A112-1-1), alanine aminotransferase (Cat. No. C009-2-1), and aspartate aminotransferase (Cat. No. C010-2-1) were obtained from the Nanjing Jiancheng Bioengineering Institute (Nanjing, China). Pomelo peel was obtained from Rongxian County, Guangxi Province, China and served as a source of naringin.

### 2.2. Naringin Preparation

The yellow exocarp of the fresh pomelo peel was dried at a constant temperature (60 °C) and humidity in a drying box and passed through a 60–mesh sieve to obtain a pomelo peel powder. The ultrasonic-assisted extraction method was used to prepare exactly 1.0 g citrus residue powder. The ethanol volume fraction was 75%. The solid/liquid ratio was 1 g/mL:55 g/mL and the extraction time, temperature, and frequency were 90 min, 75 °C, and 40 kHz, respectively. The samples were extracted under the aforementioned optimum conditions, purified with DM101 macroporous resin (China, Beijing), concentrated to approximately 1/4 of the volume of the stock solution in a vacuum, cooled and allowed to stand at 4 °C for 12 h. The precipitated light-yellow crystals were dissolved in a small volume of distilled water in a water bath maintained at 50 °C, then filtered while hot, and vacuum-dried at 50 °C to obtain a refined product of naringin [10]; 10 mg of both naringin-refined product and standard product were weighed accurately and dissolved in methanol and the volume was fixed at 100 mL. They were diluted with the same and qualitatively analyzed using ultraviolet (UV) spectrophotometry, while the purity was determined by high-performance liquid chromatography (HPLC). The conditions of HPLC were as follows: the injection amount was 5 µL; the detection wavelength was 282 nm; the Zorbax sb-c18 reversed-phase column (4.6 × 150 mm; 5 um) of the Agilent company was selected and separated the sample; the mobile phase was methanol and water (7:3); and the flow rate was 1.0 mL/min.

A Nicolet iS5 Fourier infrared spectrometer (Thermo Fisher Scientific, Waltham, Massachusetts, USA) was used to detect the samples by infrared spectroscopy (FT-IR). The potassium bromide tableting method was used by mixing the sample in potassium bromide evenly, and grinding the tableting in agate mortar, with the following scanning range: 500–4000 cm^−1^.

The refined product was analyzed via mass spectrometry (MS; ion source ESI source; methanol solvent; direct injection method; Q ExactiveTM; Thermo Fisher Scientific, Waltham, MA, USA) and nuclear magnetic resonance (NMR; deuterated acetone solvent; tetramethylsilane internal standard; AVANCE NEO 600M; Bruker Corp., Billerica, MA, USA) and compared against literature values to confirm that it was indeed pure naringin [11,12].

### 2.3. Ethical Considerations

Ethical approval for the study protocol was obtained from the Animal Care and Ethics Committee (ACEC) of Jinzhou Medical University (No. 2020041). Based on ACEC recommendations, we strove to minimize animal pain and suffering.

### 2.4. Development of Experimental Animal Model for Hyperlipidemia 

Male SPF Kunming (KM) mice (each weighing 25 ± 5 g), procured from the Experimental Animal Center of Jinzhou Medical University, Jinzhou, China (Laboratory Animal Production License Certificate No. SCXK (Liao) 2019-0003) (Laboratory Animal Use Permit Certificate No. SYXK (Liao) 2019-0007), were used in this study. Basic feed was provided by Beijing Keao Xieli Feed Co. Ltd., Beijing, China. The high-fat diet comprised of 77% basic feed, 11.5% lard, 2.3% cholesterol, 7.7% egg yolk, and 1.5% sodium deoxycholate. Each group was intragastrically administered naringin once each morning. The doses were 50 mg/kg-d (low-dose group), 100 mg/kg-d (medium-dose group), and 200 mg/kg-d (high-dose group). In the abovementioned works, the administration of 200 mg/day naringin was safe and effective [13]. The animal room was maintained at a temperature of 25 ± 3 °C and 60–70% RH.

The assays were conducted with reference to the “Test Methods and Operating Procedures of Health Food Auxiliary Hypolipidemic Function” of the Ministry of Health. Healthy KM mice were adaptively fed for 1 week and then randomly divided by body weight into two groups. The blank control group was fed a basic diet. The model group was fed a high-fat diet for 4 weeks and weighed once weekly. The normal control (NC) mice fasted for 16 h; thereafter, blood samples were drawn from the eyeballs. 

### 2.5. Model Grouping and Treatment Administration

Following hyperlipidemia model establishment, the model group was randomly divided into hyperlipidemia model groups (HM) by blood lipid levels. These consisted of the low-dose (LD), medium-dose (MD), and high-dose (HD) groups, where all mice were fed a high-fat diet as previously mentioned [14].

### 2.6. Measurement of Blood Lipid Reduction Indices

#### 2.6.1. Measurement of Mouse Body Weights and Organ Coefficients

The mice were weighed once weekly. The organ coefficient is the percentage of the ratio of visceral fresh weight to body weight and reflects total nutritional status and visceral lesions [15]. The mice were euthanized by dislocation of cervical vertebrae. Their livers, spleens, brains, hearts, and kidneys were excised. The organs were immediately weighed and stored at −80 °C until use.

#### 2.6.2. Serum Lipid Determination

Blood samples were stored in a refrigerator at 4 °C for 20 min and centrifuged at 4 °C and 4000× *g* for 10 min to obtain sera. HDL-C, LDL-C, TG, TC, ALT, and ALS levels were measured according to the kit manufacturer’s instructions (Nanjing Jiancheng Bio-Engineering Institute, Nanjing, China).

#### 2.6.3. Serum Lipid Peroxidation Index (malondialdehyde; MDA) Determination

Fresh serum (0.1 mL) was mixed with the reagent according to the kit’s instructions. The tube was tightly sealed with a film, which was then perforated with a needle. The tube was heated to 95 °C in a water bath for 40 min, cooled in water, and centrifuged at 4000× *g* for 10 min. The supernatant was collected and its OD_532_ was measured in a full-wavelength microplate reader (ReadMax1900; Shanghai Shenpu Biotechnology Co., Ltd., Shanghai, China). The standard tube contained 10 μmol mL-1 tetraethoxypropane instead of serum. Distilled water was used to zero the instrument, and the absorbance of each tube was measured [16]. 

Fresh serum (0.02 mL) was mixed with the reagent according to the kit’s instructions and incubated at a constant temperature of 37 °C in a water bath for 40 min. The developer was then added and the OD_550_ was recorded after 10 min [17]. The SOD activity was determined from the absorbance.

#### 2.6.4. Serum Glutathione Peroxidase (GSH-Px) Vitality Determination

The assay was performed according to the kit’s instructions. Supernatant (2.5 mL) absorbance was measured at 412 nm. The light path was 1 cm, and the instrument was zeroed with distilled water. The absorbance values were measured for each serum tube [18]. GSH-Px activity was calculated as follows:GSH-Px activity = (non-enzymatic absorbance)/(blank [standard] absorbance) × standard concentration (20 μmol L^−1^) × dilution ratio × sample pre-test dilution ratio,(1)

### 2.7. H and E Staining of Liver Pathological Sections

The thickness and width of the liver specimens were ≤0.3 cm and ≤0.7 cm, respectively. The specimens were placed in a container in sequence and rinsed with running water for ~24 h (Ultrathin Slicer Model EMUC6; Leica Microsystems AG, Wetzlar, Germany).The specimens were then placed in an alcohol concentration gradient of 40% (*v*/*v*), 60% (*v*/*v*), 85% (*v*/*v*), 95%( *v*/*v*) (2×), and 100% (2×) for 5 h, 5 h, 12 h, 1.5 h, and 1 h, respectively, and blotted with absorbent paper.The specimens were immersed in xylene for 15–25 min until they turned yellow and became transparent to varying degrees.The specimens were placed in a cup containing wax. The cup was heated to 56–58 °C, and the specimens absorbed the paraffin for 30 min. They were then removed and cooled.The paraffin-impregnated specimens were placed in an incubator, the melted wax was poured into a copper mold, and the specimens were then placed into it in sequence. A ≥ 1 cm distance was maintained between the specimens. After the paraffin hardened, the specimens were separated from it.The paraffin blocks were sorted by treatment group, stored at −18 °C for ≥2 h, and placed on ice blocks.Each wax block was affixed to a microtome frame (40–80 μm). The target section thickness was 5 μm. Each section was removed using tweezers, immersed in cold water, lifted with a microscope slide, immersed in water at 46 °C to spread the tissue over the slide, set on a shelf, and heated in an oven at 37 °C for ≥ 3 h.The slides were immersed in xylene (2×), 100% alcohol (2×), 95% (*v*/*v*) alcohol (2×), 85% (*v*/*v*) alcohol, 75% (*v*/*v*) ethanol, and water for 5 min each, with the exception of xylene which was immersed for 15 min. The slides were then removed, dried, and immersed in Carazzi’s modified hematoxylin for 3 min, rinsed thrice with water, subjected to 1% (*v*/*v*) HCl for 3–4 s, and immersed in water for 0.5–1 h. The slides were then placed in eosin dye, 85% (*v*/*v*) alcohol, 95% (*v*/*v*) alcohol (2×), 100% alcohol (2×), and xylene for 5 min, 2 min, 1 min, 1 min, and 1 min, respectively. Then, they were removed and dried, and coverslips were sealed onto them with an adhesive [19].

### 2.8. Acute Toxicity Analysis of Citrus Grandis

The maximum naringin dose was orally administered to the mice in this study. Mortality rates and acute naringin toxicity were determined based on an acute toxicity test (National Food Safety Standard Acute Oral Toxicity Test No. GB15193.3-2014). Twenty mice were randomly and equally divided into the blank and treatment groups. The mice were fasted for 6 h before the assay, and they had ad libitum water access. The mice in the treatment groups were orally administered 400 mg/kg naringin at 8 h intervals in 1 d. The blank group was administered an equivalent volume of distilled water for 21 days. Behavior, food intake, activity level, and weight were monitored [20].

### 2.9. Reverse Transcription-Quantitative PCR (RT-qPCR)

Perfectstart®, Beijing, China and the green qPCR Supermix kit, Quanshijin Biotechnology Co., Ltd, Beijing, China were used for the fluorescent RT-qPCR. The dry primer powders were reconstituted according to the primer’s instructions and mixed via centrifugation. Each reaction was repeated thrice, and a negative control was set. Pre-mix cDNA Quanshijin Biotechnology Co., Ltd., Beijing, China, primer Quanshijin Biotechnology Co., Ltd., Beijing, China, RNase free water Solabo Technology Co., Ltd., Beijing, China, and SYBR Quanshijin Biotechnology Co., Ltd., Beijing, China, were prepared on ice in advance. The reaction system is shown in Appendix A. The pre-mixed reaction system was added to the enzyme removal 96-well plate, and the plate was sealed with a film. The contents of the plate were through centrifugation, and the plate was placed in a real-time fluorescent quantitative PCR instrument for detection. The “two-step method” was adopted for detection: the first step involved 95 °C pre-denaturation for 3 min; the second step involved denaturation at 92 °C for 10 s, followed by 40 cycles of annealing and extension for 30 s. The column chart indicated the ratio of the expression of the target gene to that of the internal reference gene (β-actin). β-actin was used as a reference in the qRT-PCR. The primers used are listed in Appendix A.

### 2.10. Statistical Analysis

All data analyses were performed in triplicate. Data were expressed as means ± standard error (±SE). *p* < 0.05 and *p* < 0.01 indicated significant and very significant differences between group means, respectively. SPSS v. 22.0 (IBM Corp., Armonk, NY, USA) was used for the ANOVA and Duncan’s multiple range comparison tests [21]. Data were also analyzed and plotted with GraphPad Prism v. 5.01 (GraphPad Software, La Jolla, CA, USA).

## 3. Results and Discussion

### 3.1. Qualitative and Quantitative Analysis of Naringin

The UV, IR, and NMR (Appendix A) of our refined naringin products were shown in Appendix A and compared with the standard naringin or standard library [22]. The results showed that the spectrogram of the separated naringin was highly consistent with that of the standard. The m/z value of refined naringin products was in accordance with the molecular weight of the standard naringin, which further proved that the refined products were naringin. The UV, IR, and NMR spectra of naringin were also assigned in the Appendix A.

The purity of naringin in our refined naringin products was determined via HPLC, after the refined and standard naringin products were preliminarily determined as the same substance by UV spectroscopy. The same amounts of refined and standard naringin (purity 98%) were passed through the HPLC column under the same conditions to obtain a chromatogram.

The chromatographic peak type and retention time of both the refined and standard products were approximately the same (Figure 1), which indicates a high purity of naringin in our refined products. According to the ratio of the peak area of the refined and standard products, the purity ratio of both was calculated, and finally the content of naringin in our refined, experimental product was obtained. The purity was calculated to be 95.46%.

### 3.2. Hyperlipidemia Model Establishment

The model mouse group was fed a high-fat diet for 4 weeks, and its TG, TC, HDL-C, and LDL-C levels were measured. The results are shown in Table 1. Compared with the blank group, the TC level in the model group increased by 2.8 times, the TG level increased by 1.6 times, the LDL-C level increased by 2 times, and the HDL level in the model group decreased by 87.1% (*p* < 0.01), indicating that the high-fat model was successfully established.

### 3.3. Effects of Naringin on Mouse Body Weight

Figure 2A shows that after 35 days of feeding, the body weight of mice in each group increased steadily. The body weight of mice in the blank control group, model group, low-dose group, medium-dose group, and high-dose group increased by 22.23%, 32.38%, 39.75%, 29.21%, and 25.39%, respectively. From the third week, the body weight growth of mice in the high-dose group and medium-dose group slowed down. Compared with the model group, the high-dose group and the medium-dose group had significant differences (*p* < 0.01), indicating that naringin could inhibit the weight gain caused by a high-fat diet in a dose-dependent manner.

### 3.4. Effects of Naringin on Mouse Blood Lipid Levels

Figure 2B shows that compared with the blank group, the levels of TC, TG, and LDL-C in the model group increased significantly (*p* < 0.01), and the level of HDL-C decreased significantly (*p* < 0.01). After the intragastric administration of naringin for 5 weeks, compared with the model group, the TC level of mice in low-, medium-, and high-dose groups decreased by 25.80%, 35.73%, and 40.01%; the TG level decreased by 26.81%, 44.16%, and 61.20%; the LDL-C level decreased by 10.92%, 18.77%, and 27.30% (*p* < 0.01); and the HDL-C level increased by 3.44%, 5.17%, and 5.66% (*p* < 0.01), in a dose-dependent manner. Therefore, high naringin doses significantly lowered blood lipids in mice fed a high-fat diet.

Hyperlipidemia is a metabolic disease characterized by abnormal lipid metabolism and elevated serum low-density lipoprotein cholesterol (LDL-C), triglyceride (TG), total cholesterol (TC), and depressed high-density lipoprotein cholesterol (HDL-C) levels [4]. Therefore, it is vital to develop natural plant functional factors and drugs that safely and effectively lower blood lipid levels [23]. The present study demonstrated the ability of naringin to reduce blood lipid levels and significantly inhibit weight gain in mice. Naringin significantly lowered serum TC and TG levels in hyperlipidemic mice and prevented the subsequent increase in the levels of these blood lipid markers. Therefore, naringin could promote cholesterol metabolism, prevent arterial and hepatic lipid deposition, and lower blood lipid levels. Cholesterol per se is fat-soluble. It only becomes soluble in the blood after binding to lipoproteins. HDL-C helps remove the “bad” cholesterol deposited in blood vessel walls and unclogs the blood vessels. By contrast, LDL-C readily adheres to blood vessel walls and increases blood lipid levels. The results of this study showed that high naringin doses significantly inhibited the reduction in serum HDL-C levels and the increase in serum LDL-C levels in hyperlipidemic mice. Naringin participates in cholesterol metabolism and prevents lipid deposition on blood vessel walls. Naringin somewhat influenced mouse ALT and ALS activity. The high-dose, medium-dose, and low-dose treatments reduced serum lipids to levels below those of the blank control.

### 3.5. Effects of Naringin on Mouse Serum SOD and MDA Levels

Figure 2C shows that compared with the model group, the level of MDA in the liver of high-dose mice decreased by 35.65%, the level of MDA in the liver of medium-dose mice decreased by 20.61%, and the level of MDA in the liver of low-dose mice decreased by 14.03% (*p* < 0.01). Figure 2D shows that the level of SOD in the liver of high-dose mice increased by 45.51%, and the level of SOD in the liver of medium-dose mice increased by 24.43%. The level of SOD in the liver of low-dose mice increased by 15.70% (*p* < 0.01). Therefore, naringin can inhibit lipid peroxidation and improve the anti-oxidant capacity of liver tissue, and the effect of the high-dose group was the best.

Malondialdehyde (MDA) is a major by-product of lipid peroxidation. MDA levels directly and indirectly reflect the degrees of lipid peroxidation and cell damage, respectively. GSH-Px and SOD are vital antioxidant enzymes. The former catalyzes the reduction of GSH to H_2_O_2_. GSH is a ubiquitous bioactive peptide and a cofactor in various metabolic pathways. It participates in the metabolism of numerous cellular compounds and removes free radicals generated by drug and toxin metabolism. SOD helps balance oxidation and the anti-oxidant system by removing superoxide anion free radicals and protecting cells from oxidative damage. SOD activity indirectly reflects the ability of an organism to scavenge free radicals. Here, naringin significantly upregulated serum GSH-Px and SOD and lowered serum MDA in mice fed a high-fat diet.

### 3.6. Effects of Naringin on Mouse Liver Histopathology

The histopathological analyses of mouse liver are shown in Figure 3A–E. As is shown in the blank group, the hepatocyte cord structure was clear, the hepatocyte structure was normal, and there were no obvious pathological changes. In the model group, the hepatocyte cords were disorganized, the hepatic lobule structure was unclear, the hepatocytes contained vacuoles of various sizes, diffuse steatosis was present, some hepatocytes had degenerated and were necrotic, and their nuclei had shrunk and decomposed. In the high-dose group, the hepatic lobule and hepatocyte cord structures were normal, the hepatocyte cytoplasm stained red, and there were no obvious pathological changes. In the medium-dose group, the hepatocyte cord structure was basically normal, a few small vacuoles appeared in the hepatocyte cytoplasm, and the lesions were less severe than those of the low-dose group. In the low-dose group, the hepatocyte cord structure was basically normal, small-to-medium vacuoles appeared in certain hepatocyte cytoplasm, and there were moderate lesions outward from the central vein. The foregoing sections were examined under a light microscope at 400x (IX73P1F; Olympus, Tokyo, Japan).

### 3.7. Acute Naringin Toxicity Assay

On the same day of intragastric naringin administration, mouse appetites slightly declined. After 3 h, mouse feeding and activity returned to normal. The mice were observed for 21 days, during which they presented with no toxicity symptoms, and there was no mortality. The mice were euthanized on day 21. The autopsy revealed no obvious changes in any internal organ or tissue. Mouse weight changes are shown in Table 2. For all naringin treatments, the weight gain was in the normal growth range. Compared with that in the blank group, there was no remarkable difference in the body weight of the naringin-treatment group. Therefore, the acute toxicity test established that naringin was non-toxic.

### 3.8. Effects of Naringin on Mouse Organ Indices

The effects of naringin on the organ indices of hyperlipidemic mice are shown in Figure 2E.

The liver indexes of the blank group and the model group were 2.70 and 5.86, respectively, and there was a significant difference between the two groups (*p* < 0.01). With an increase in the naringin intragastric dose, the liver index gradually decreased, indicating that there was a negative correlation between the naringin dose and liver index. The liver indices of the naringin low-, medium-, and high-dose groups were 4.30, 3.79, and 3.58, respectively, which were significantly different from those of the model group (*p* < 0.01), indicating that naringin has a certain nutritional improvement effect on the livers of mice. Compared with the blank control group, there was no significant difference in the indexes of the brain, heart, spleen, and kidney between the low- and medium-dose groups. Therefore, it can be preliminarily inferred that the low- and medium-dose naringin will not cause actual damage to the organs of mice.

### 3.9. Effects of Naringin on Mouse Liver Function

Figure 2F shows that compared with those in the blank group, ALT levels increased in the model group, low-dose, medium-dose, and high-dose groups, and the difference was significant (*p* < 0.01). Compared with those in the model group, ALT levels decreased by 20.55%, 38.96%, and 46.77% in the low-dose, medium-dose, and high-dose groups, respectively (*p* < 0.01). As can be seen from Figure 2G, compared with those in the blank group, AST levels increased in the model, low-dose, medium-dose, and high-dose groups, and the difference was significant (*p* < 0.01). Compared with those in the model group, AST levels decreased by 3.99%, 36.38%, and 48.42% in the low-dose, medium-dose, and high-dose groups, respectively (*p* < 0.01). In conclusion, naringin can protect the livers of mice fed a high-fat diet. 

Hyperlipidemia strongly affects the mouse liver and other organs [24,25]. The heart, kidney, liver, spleen, and brain indices of hyperlipidemic mice were significantly higher than those of the blank control mice. Naringin administration did not significantly affect the mouse brain but improved apparent nutrient storage and metabolism in the mouse liver. Compared with those in the blank control, there were no significant differences in the brain, heart, spleen, or kidney indices of the medium-dose and low-dose groups. These discoveries suggest that neither low-dose nor medium-dose naringin damages mouse organs.

### 3.10. Effects of Naringin on GSH-Px Activity

The effects of naringin on serum GSH-Px activity are shown in Figure 2H. As can be seen from Figure 2H, the GSH-Px activities of the blank group and the model group were 36.91 µmol/L and 26.81 µmol/L, respectively, and the difference between the two groups was significant (*p* < 0.01). With an increase in the dose, the activity of GSH-Px gradually increased, indicating that the activity of GSH-Px was dose-dependent. The activities of low, medium, and high doses of naringin GSH-Px were 27.23 µmol/L, 31.81 µmol/L, and 34.58 µmol/L, respectively, compared with those in the model group; there was no significant difference in the activity of GSH-Px in the low-dose group, while there was a significant difference in the activity of GSH-Px in the medium-dose and high-dose groups (*p* < 0.01), indicating that naringin can effectively improve the activity of GSH-Px in the serum. In this study, the activity of serum GSH-Px in the high-dose group was significantly higher than that in the blank control group (*p* < 0.01).

### 3.11. Effect of Naringin on FAS mRNA Expression in Mouse Liver

The results of FAS mRNA expression in liver tissue are shown in Figure 4A. Compared with the blank control group, the expression of FAS mRNA in liver tissue of low-, medium-, and high-dose groups was significantly different (*p* < 0.01). Compared with the blank control group, the expression of FAS mRNA in the low-, medium-, and high-dose groups was increased by 42.97%, 15.59%, and 5.87%, respectively. Compared with the model group, the FAS mRNA expression in the liver tissue of the low-, medium-, and high-dose groups was significantly different (*p* < 0.01), and decreased by 9.46%, 26.79%, and 32.95%, respectively.

Histopathological analyses of the mouse liver showed that the hepatocyte cords were disorganized, the hepatic lobule structures were unclear, the hepatocytes contained vacuoles of various sizes, diffuse steatosis was present, some hepatocytes had degenerated and were necrotic, and their nuclei shrank and decomposed in the model group. Compared with the model group, all naringin-treatment groups presented with normal hepatic lobule structures and hepatocyte cords, red-stained hepatocyte cytoplasm, and no inflammatory cell infiltration. Moreover, liver amelioration appeared to be dose-dependent. This suggests that naringin may mitigate fatty hepatocyte degeneration injury in hyperlipidemic mice. Furthermore, the acute toxicity test revealed that naringin was safe, efficacious, and non-toxic in mice.

The liver maintains the main metabolic function in the body. The liver participates in the storage of liver glycogen, synthetic albumin, vitamin metabolism, bile production, and excretion. Furthermore, it is also an important site for the synthesis of fatty acids and fats, and it is the most vigorous organ for the synthesis of cholesterol in the human body. The liver-synthesized cholesterol accounts for more than 80% of the total amount of synthetic cholesterol in the body and is the main source of plasma cholesterol. High-fat diets can cause the accumulation of lipids in liver tissue, along with high FAS expression in the liver tissues of obese patients. Consequently, FAS has become a new drug target for the study of such diseases. The study of FAS inhibitors is of great significance for inhibiting the biosynthesis of endogenous fatty acids and effectively controlling the occurrence and development of tumors, obesity, and various related metabolic syndromes. Chen B et al. (2012) discussed the effects of purslane water extract on blood lipid, oxidative stress, and par in the livers of hyperlipidemic mice and FAS mRNA. It was found that purslane water extract could not only reduce the blood lipid level and reduce oxidative stress, but also significantly inhibit the expression of FAS mRNA in the liver and spleen. In addition, flavonoids in plants can inhibit the proliferation and promote the apoptosis of human breast cancer cells by inhibiting the expression of FAS. The results showed that the expression of FAS mRNA in the liver tissue of the model group without treatment was significantly higher than that in the blank control group, indicating that a long-term high-fat diet may lead to a high expression of FAS mRNA. After five weeks of naringin treatment, the FAS mRNA expression in the liver tissue of each group was significantly lower than that in the model group, indicating that naringin can inhibit FAS mRNA expression in the liver tissue to some extent.

### 3.12. Effect of Naringin on PPARα mRNA Expression in Mouse Liver

The results of PPARα mRNA expression in liver tissue are shown in Figure 4B. Compared with that in the blank control group, there was no significant difference in the expression of PPARα mRNA in liver tissue between the low- and medium-dose groups. The expression of PPARα mRNA in liver tissues of the high-dose group was significantly different (*p* < 0.05). Compared with that in the blank group, the expression of PPARα mRNA in the high-fat model group was downregulated by 12.59%. Compared with that in the model group, the expression of PPARα mRNA in liver tissues of the low-, medium-, and high-dose groups was significantly higher (*p* < 0.01) and was upregulated by 3.05%, 9.68%, and 41.73%, respectively.

PPARα can reduce blood triglyceride levels. The results of this experiment showed that the expression of PPARα mRNA in the liver tissue of the model group without treatment was lower than that in the blank control group, indicating that a long-term high-fat diet may lead to a low expression of PPARα mRNA. After five weeks of naringin treatment, the expression of PPARα mRNA in the liver tissue of each group was significantly higher than that in the model group, indicating that naringin could improve the expression of PPARα mRNA in liver tissue to a certain extent.

### 3.13. Effect of Naringin on SREBP-1c mRNA Expression in Mouse Liver

The results of SREBP-1c mRNA expression in liver tissue are shown in Figure 4C. Compared with that in the blank control group, the expression of SREBP-1c mRNA in liver tissues of the low-, medium-, and high-dose groups was significantly different (*p* < 0.01). Compared with that in the blank control group, SREBP-1c mRNA expression was upregulated by 85.71%, 80.63%, and 74.58%, in the liver tissues of the low-, medium-, and high-dose groups, respectively. Compared with that in the model group, the expression of SREBP-1c mRNA was significantly different (*p* < 0.01) and downregulated by 20.27%, 22.46%, and 25.08% in the liver tissues of the low-, medium-, and high-dose groups, respectively.

SREBP-1c is another important transcription factor that regulates lipid homeostasis by regulating cholesterol homeostasis, fatty acid synthesis, and TG-related gene expression. SREBP-1c preferentially activates fatty acid and TG metabolism genes [26]. Fatty acids, as raw materials for TG synthesis, are closely related to TG synthesis. FAS directly catalyzes the biosynthesis of fatty acids and is regulated by SREBP-1c. The results of our study indicated that the expression of SREBP-1c mRNA in the liver tissue of the model group without treatment was significantly higher than that in the blank control group, indicating that a long-term high-fat diet may lead to a high expression of SREBP-1c mRNA. After five weeks of naringin treatment, the expression of SREBP-1c mRNA in the liver tissue of each group was significantly lower than that in the model group, indicating that naringin could inhibit the expression of SREBP-1c mRNA in the liver tissue to some extent. This further indicated that naringin could reduce serum TG levels and inhibit liver lipid accumulation.

## 4. Conclusions

Naringin is a natural active substance extracted from pomelo peel. Naringin significantly (*p* < 0.01) reduced the body weight, organ index, serum TG, LDL-C, and TC levels of hyperlipidemic mice, but significantly (*p* < 0.01) increased the serum HDL-C levels. In addition, naringin significantly (*p* < 0.01) increased GSH-Px and SOD activity; significantly (*p* < 0.01) decreased MDA, ALT, and AST levels; and significantly (*p* < 0.01) decreased the liver index. There was no statistically significant difference in the brain, heart, spleen, kidney, and other indicators (*p* > 0.05). A histopathological analysis of mouse liver showed that naringin could alleviate the degenerative damage of fatty liver cells in hyperlipidemic mice. Naringin could significantly (*p* < 0.01) reduce the expression of FAS and SREBP-1c mRNA, and significantly (*p* < 0.01) increase PPARα mRNA expression. This study shows that naringin has the strong effect of lowering lipids and protecting the liver in hyperlipidemic mice. Our findings underscore the anti-hyperlipidemia potential of naringin and increase the scientific understanding of the anti-hyperlipidemia effects of naringin, which could lead to its potential application as a dietary strategy for hyperlipidemia management in the future.

## Figures and Tables

**Figure 1 molecules-27-09033-f001:**
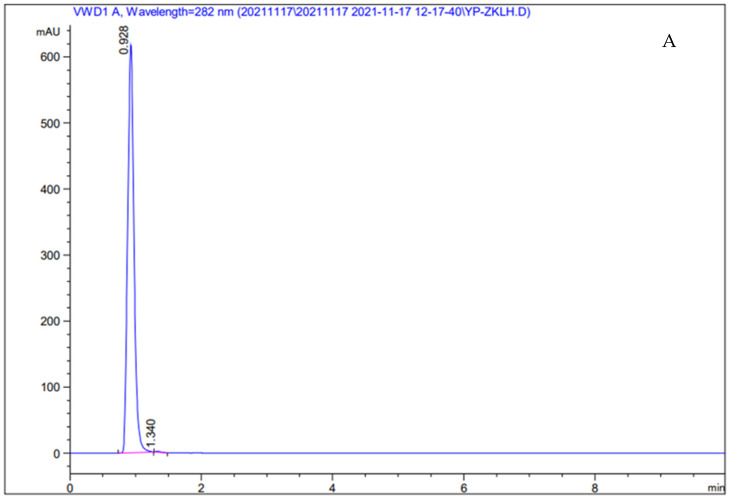
HPLC diagram of (**A**) naringin-refined products and (**B**) its standards (the chromatographic peak type and retention time of both the refined and standard products were approximately the same).

**Figure 2 molecules-27-09033-f002:**
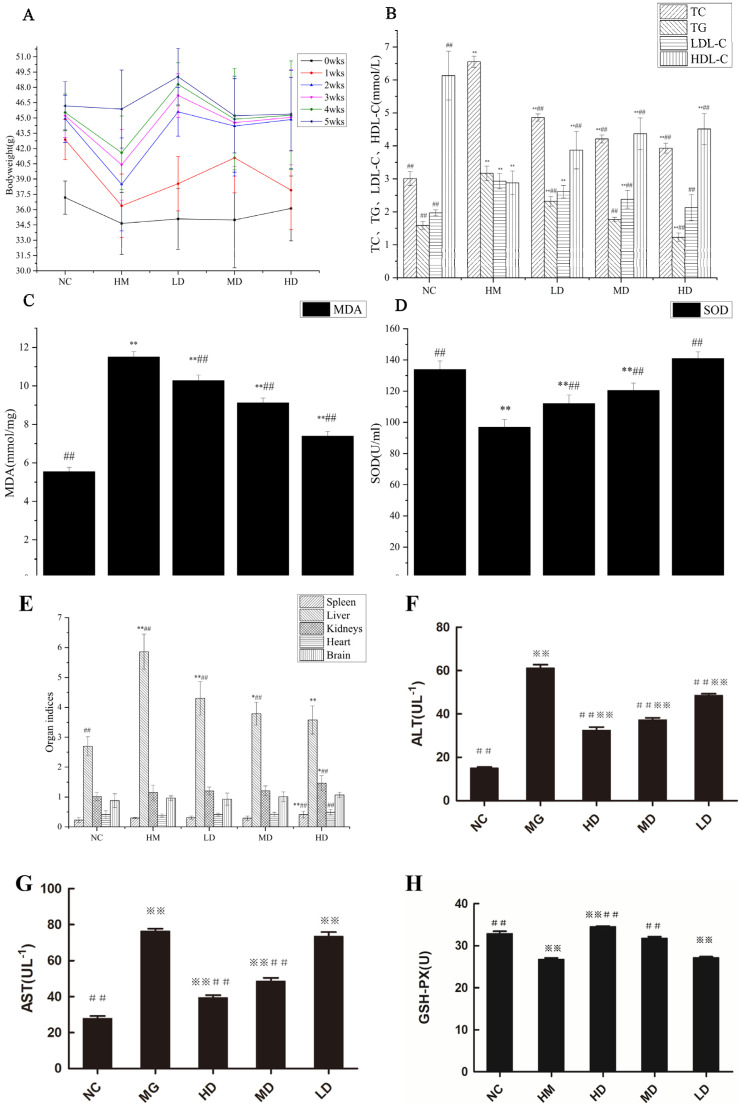
Effect of naringin on hyperlipidemic mice. Values are mean ± SD, n = 8. (**A**) Body weight. (**B**) Serum TC, TG, LDL-C, HDL-C levels. (**C**) Serum MDA levels. (**D**) Serum SOD levels. (**E**) Organ indices. (**F**) Serum ALT levels. (**G**) Serum AST levels. (**H**) Serum GSH-Px levels. Note: n = 8; * *p* < 0.05, ** *p* < 0.01 compared with those in the blank control. ## *p* < 0.01 compared with those in the model group.

**Figure 3 molecules-27-09033-f003:**
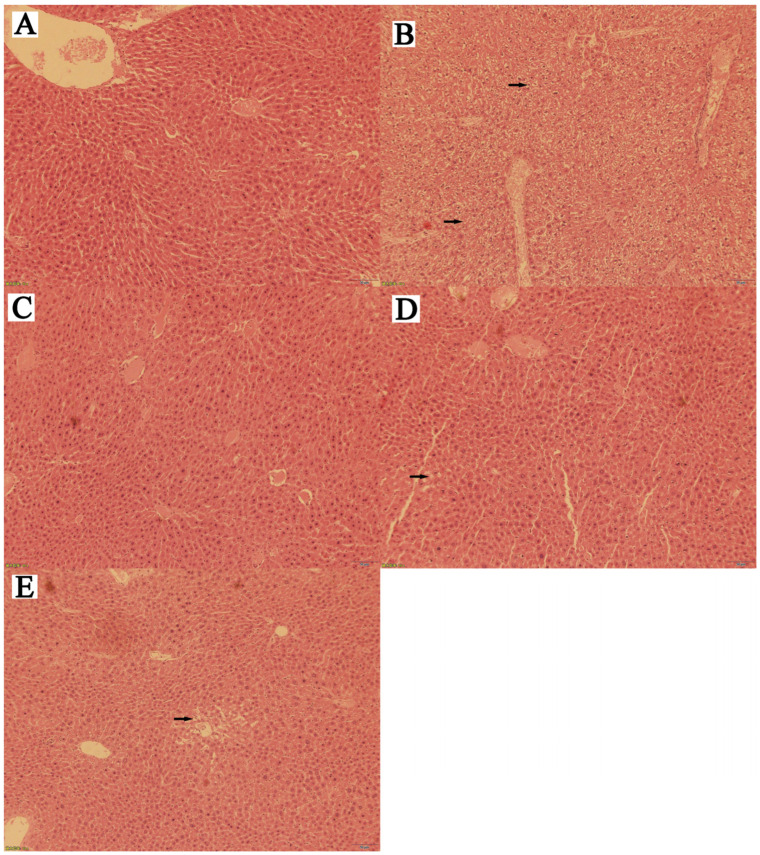
Effects of naringin on histopathological changes in mouse liver sections (the histopathological analyses of mouse liver) (H&E staining; 400×). (**A**) Normal control, NC; (**B**) hyperlipidemia model group, HM; (**C**) high-dose group, HD; (**D**) medium-dose group, MD; (**E**) low-dose group, LD.

**Figure 4 molecules-27-09033-f004:**
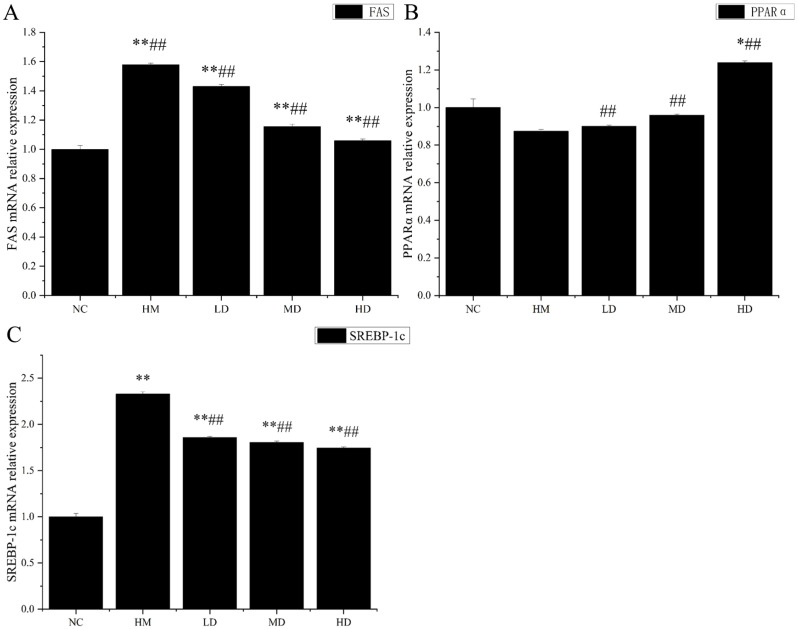
(**A**) Effects of naringin on FAS mRNA relative expression in mouse liver (the results of FAS mRNA expression in liver tissue). (**B**) Effects of naringin on PPARα mRNA expression in mouse liver (the results of PPARα mRNA expression in liver tissue). (**C**) Effects of naringin on SREBP-1c relative expression in mouse liver (the results of SREBP-1c mRNA expression in liver tissue). Note: n = 8; * *p* < 0.05, ** *p* < 0.01 compared with those in the blank control. ## *p* < 0.01 compared with those in the model group.

**Table 1 molecules-27-09033-t001:** Effects of a 28 d high-fat diet on mouse blood lipid levels (x ± SE; n = 5).

Groups	Number of Animals	TC (mmol/L)	TG (mmol/L)	LDL-C (mmol/L)	HDL-C (mmol/L)
NC	5	1.65 ± 0.77	1.43 ± 0.56	1.39 ± 0.54	5.72 ± 1.59
HM	5	6.34 ± 0.91 **	3.70 ± 0.77 **	4.18 ± 0.85 **	0.74 ± 0.09 **

Note: n = 5; ^**^
*p* < 0.01, compared with blank control.

**Table 2 molecules-27-09033-t002:** Effects of naringin on mouse body weight.

Group	Animals	Deaths	Weight (g, x ± SE)		14.d	21.d
			0.d	7.d		
Blank	10	0	20.25 ± 1.19	29.19 ± 3.11	33.55 ± 2.96	41.31 ± 2.56
Treatment	10	0	21.36 ± 1.44	28.51 ± 2.42	34.11 ± 2.66	38.52 ± 6.13

Note: significant difference between blank control and treatment groups.

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
