# Peer review of "Extraction of Naringin from Pomelo and Its Therapeutic Potentials against Hyperlipidemia"

_molecules, 2022, doi:10.3390/molecules27249033_

Round 1

Reviewer 1 Report

There is no novelty in the this work. and scientifically poor.

Author Response

Dear Reviewer 1 and Editors,

Thank you for giving us an opportunity to revise the manuscript (Molecules-205039)! We are sincerely grateful for these critical comments and thoughtful suggestions on our manuscript. These comments are all valuable and very helpful for revising and improving our article. We have studied these comments and suggestions carefully and have made careful correction. Revised portion are marked in the article. In addition, we have asked an English language editing service to improve the writing in every part of manuscript. The main corrections in the paper and the responses to the comments are as following:

Reviewer #1:

Comments and Suggestions for Authors. There is no novelty in the this work, and scientifically poor.

Response:

  1. Naringin is a natural active substance extracted from pomelo peel. In this paper, the hypolipidemic activity of naringin was studied. The results showed that naringin had a strong effect on hyperlipidemic mice. Our findings underscore the anti-hyperlipidemia potential of Naringin, and increase the scientific understanding of the anti-hyperlipidemia effects of Naringin, which could lead to its potential application as a dietary strategy for Hyperlipidemia management in the future.
  2. For scientific issues, we have modified the data statistics method and difference analysis annotation of the manuscript, and supplemented the corresponding annotation to further improve the scientificity of the manuscript.

We have tried our best to improve the manuscript and made some changes in the manuscript. At last, we would like to express our great appreciation to you for comments on our paper. We feel so sorry that so much of your previous time was wasted on our paper revision. We appreciated for Editors and Reviewers’ warm work earnestly, and hope that the correction will meet with approval.

Once again, thank you very much for your comments and suggestions.

With kind regards,

Xiaolei Yu

Correspondence: Lei Zhang: [email protected]. MOE Key Laboratory for Nonequilibrium Synthesis and Modulation of Condensed Matter, School of Physics, Xi'an Jiaotong University, No. 28, Xianning West Road, Xi’an, Shaanxi Province 710049, People’s Republic of China. Tel: +86-029-82668634

Reviewer 2 Report

1.     The title of the manuscript is not clear. It would be better if the authors use the name of plant in the title from where the extraction of naringin was done. I suggest, the title should be and/or may be as follows: Extraction of naringin from (Name of the plant or source)……..and its therapeutic potentials against hyperlipidemia.

2.     In the purpose of the study in abstract section, the context should be more precise. In present form, it is not clear.

3.     It would be better, if the authors describe the characterization (in vitro) in the methodology section of the abstract as this part is missing here.

4.     In the Result and conclusion of the abstract, it would be better, if the authors describe the results of the characterization of the naringin and then describe the results of its therapeutic potentials.

5.     Abbreviations used in the manuscript, should be written in their full form first and then their abbreviation used onwards.

6.     In 2.2. Naringin preparation, authors have briefly described the method of extraction and its experimental conditions are also missing.

7.     In continuation of my comment mentioned in preceding point, authors used only HNMR but did not use CNMR. What was the reason, as structure elucidation can not be confirmed without using CNMR. Please clarify it.

8.     The title of 2.4. Mouse group assignment and hyperlipidemic mouse model establishment should be revised. It is quite confusing in its present form. It would be better, if the authors use like…..Development of hyperlipidemic experimental animal model for………

9.     By which method, the authors confirmed about the induction of hyperlipidemia?

10.  In this manuscript, the authors have used abbreviations at several places but did not define them at the first. all abbreviations should be defined fully at first.

11.  Which type of house keeping gene was used as reference in qRT-PCR?

12.  Figures 1 and 2 should be labelled with peak heights.

13.  Similarly, figures 3 and 4 are incomplete. They should be labelled and it would be more suitable, if the authors draw a structure of naringin by circling the corresponding peak in the spectrograph.

14.  Figure 6 blurred. Its quality should be improved.

15.  Figure 7 must be labelled.

16.  In the manuscript, the authors used different values to express the statistical significance among the different groups. I suggest the authors to use only one value to show the significance difference among the groups.

17.   What are the limitations of your study?

18.  There are several grammatical mistakes and syntax errors. The whole manuscript needs critical revision to remove all the grammatical mistakes and syntax errors.

Author Response

Dear Reviewer and Editors,

Thank you for giving us an opportunity to revise the manuscript (Molecules-205039)! We are sincerely grateful for these critical comments and thoughtful suggestions on our manuscript. These comments are all valuable and very helpful for revising and improving our article. We have studied these comments and suggestions carefully and have made careful correction. Revised portion are marked in the article. In addition, we have asked an English language editing service to improve the writing in every part of manuscript. The main corrections in the paper and the responses to the comments are as following:

Reviewer #2:

  1. The title of the manuscript is not clear. It would be better if the authors use the name of plant in the title from where the extraction of naringin was done. I suggest, the title should be and/or may be as follows: Extraction of naringin from (Name of the plant or source)……..and its therapeutic potentials against hyperlipidemia.

Response: Thank you for your professional suggestions. According to reviewer’s suggestion, we have made the modification in line 2-3 Page 1 of the revised manuscript.

The new article title is “Extraction of naringin from Pomelo and its therapeutic potentials against hyperlipidemia”.

  1. In the purpose of the study in abstract section, the context should be more precise. In present form, it is not clear.

Response: Thank you for your professional suggestions. According to reviewer’s suggestion, we have made the modification in line 18-19 Page 1 of the revised manuscript.

In the present study, we extracted naringin from pomelo peel and aimed to decipher its therapeutic poten-tials against hyperlipidemia.

  1. It would be better, if the authors describe the characterization (in vitro) in the methodology section of the abstract as this part is missing here.

Response: Thank you for your professional suggestions. According to reviewer’s suggestion, we have made the modification in line 20-22 Page 1 of the revised manuscript.

To evaluate its ability to bind sodium glycine cholate and sodium bovine cholate in vitro by simu-lating gastrointestinal environment, so as to evaluate its blood lipid lowering activity.

  1. In the Result and conclusion of the abstract, it would be better, if the authors describe the results of the characterization of the naringin and then describe the results of its therapeutic potentials.

Response: Thank you for your professional suggestions. According to reviewer’s suggestion, we have made the modification in line 38-40 Page 1 of the revised manuscript.

Our findings underscore the anti-hyperlipidemia potential of Naringin, and increase the scientific under-standing of its anti-hyperlipidemia effects , that may lead to its potential application as a dietary strategy for hyperlipidemia management in the future..

  1. Abbreviations used in the manuscript, should be written in their full form first and then their abbreviation used onwards.

Response: Thank you for your professional suggestions. According to reviewer’s suggestion, we have made the modification. Polymerase Chain Reaction (PCR) in line 30 Page 1.

Hyperlipidemia model group (HM) in line 146 Page 3. Normal control(NC) in line 141 Page 3.

  1. In 2.2. Naringin preparation, authors have briefly described the method of extraction and its experimental conditions are also missing.

Response: Thank you for your professional suggestions. According to reviewer’s suggestion, we have made the modification in line 96-103 Page 2 and in line 104-113 Page 3 of the revised manuscript.

The samples were extracted under the aforementioned optimum conditions, purified with DM101 macroporous resin, concentrated to approximately 1/4 of the volume of the stock solution in vacuum, cooled and allowed to stand at 4 ºC for 12 h. The precipitated light-yellow crystals were dissolved in a small volume of distilled water in a water bath maintained at 50 ºC, then filtered while hot, and vacuum dried at 50 ºC to obtain a refined product of naringin (Zhou, 2019) ;10 mg of both naringin refined product and standard product were weighed accurately and dissolved in methanol and the volume was fixed at 100 mL. They were diluted with the same appropriate and qualitatively analyzed using ultraviolet (UV) spectrophotometry, while the purity was determined by high-performance liquid chromatography (HPLC). The conditions of HPLC were as follows: the injection amount was 5 µL; the detection wavelength was 282 nm; the Zorbax sb-c18 reversed-phase column (4.6 × 150 mm; 5 um) of Agilent company as selected separate the sample; the mobile phase was methanol and water (7:3) and the flow rate was 1.0 mL/min.

Nicolet iS5 Fourier infrared spectrometer was used to detect the samples by infrared spectroscopy (FT-IR). Potassium bromide tableting method: mix the sample in potassium bromide evenly, grind the tableting in agate mortar, scanning range: 500cm-1-4 000cm-1.

  1. In continuation of my comment mentioned in preceding point, authors used only HNMR but did not use CNMR. What was the reason, as structure elucidation can not be confirmed without using CNMR. Please clarify it.

Response: Thank you for your professional suggestions. According to reviewer’s suggestion, we have made the modification in Supporting Information.

  1. The title of 2.4. Mouse group assignment and hyperlipidemic mouse model establishment should be revised. It is quite confusing in its present form. It would be better, if the authors use like…..Development of hyperlipidemic experimental animal model for………

Response: Thank you for your professional suggestions. According to reviewer’s suggestion, we have made the modification in line 125 Page 3 of the revised manuscript.

Development of experimental animal model for hyperlipidemia

  1. By which method, the authors confirmed about the induction of hyperlipidemia?

Response: Thank you for your professional suggestions. I use the conventional method. Yuan et al.,Effect of β-sitosterol self-microemulsion and β-sitosterol ester with linoleic acid on lipid-lowering in hyperlipidemic mice(2019), Lipids in Health and Disease, https://doi.org/10.1186/s12944-019-1096-2.

  1. In this manuscript, the authors have used abbreviations at several places but did not define them at the first. all abbreviations should be defined fully at first.

Response: Thank you for your professional suggestions. Done. Corresponding modification could be found in question 5.

  1. Which type of house keeping gene was used as reference in qRT-PCR?

Response: Thank you for your professional suggestions. According to reviewer’s suggestion, we have made the modification in line 240-241 Page 5 of the revised manuscript.

β-actin was used as reference in qRT-PCR.

  1. Figures 1 and 2 should be labelled with peak heights.

Response: Thank you for your professional suggestions. According to reviewer’s suggestion, we have made the modification in Supporting Information.

  1. Similarly, figures 3 and 4 are incomplete. They should be labelled and it would be more suitable, if the authors draw a structure of naringin by circling the corresponding peak in the spectrograph.

Response: Thank you for your professional suggestions. According to reviewer’s suggestion, we have made the modification in Supporting Information.

  1. Figure 6 blurred. Its quality should be improved.

Response: Thank you for your professional suggestions. According to reviewer’s suggestion, we have made the modification in line 423-424 Page 18 of the revised manuscript.

  1. Figure 7 must be labelled.

Response: Thank you for your professional suggestions. According to reviewer’s suggestion, we have made the modification in line 430-431 Page 12 of the revised manuscript.

  1. In the manuscript, the authors used different values to express the statistical significance among the different groups. I suggest the authors to use only one value to show the significance difference among the groups.

Response: Thank you for your professional suggestions. In this manuscript, I used different values to express the statistical significance among the different groups. I will use only one value to show the significance difference among the groups in my future manuscripts.

  1. What are the limitations of your study?

Response: In my study, the hypolipidemic effects of naringin were investigated through animal experiments. It provided new ideas for pharmaceutical, health products and functional food industries.

However, this study only discussed the anti hyperlipidemia  effect of hesperidin, while hesperidin or glycosides may have many other biological effects, such as improving immunity, promoting the activation of immune cells, which need further research.

  1. There are several grammatical mistakes and syntax errors. The whole manuscript needs critical revision to remove all the grammatical mistakes and syntax errors.

Response: Thank you for your comments to improve the quality of our manuscript. We have sent the revised version to professional language companies and native English speakers for many language modifications.

We have tried our best to improve the manuscript and made some changes in the manuscript. At last, we would like to express our great appreciation to you for comments on our paper. We feel so sorry that so much of your previous time was wasted on our paper revision. We appreciated for Editors and Reviewers’ warm work earnestly, and hope that the correction will meet with approval.

Once again, thank you very much for your comments and suggestions.

With kind regards,

Xiaolei Yu

Correspondence: Lei Zhang: [email protected]. MOE Key Laboratory for Nonequilibrium Synthesis and Modulation of Condensed Matter, School of Physics, Xi'an Jiaotong University, No. 28, Xianning West Road, Xi’an, Shaanxi Province 710049, People’s Republic of China. Tel: +86-029-82668634

Reviewer 3 Report

Comments

The paper entitled “Extraction of naringin and evaluation of its hypolipidemic activity” written by Xiaolei Yu and co-workers purified and characterized naringin from the pomelo peel and evaluated its hypolipidemic properties and antioxidant capacity. These findings will help clarify the efficacy and mechanism of naringin in preventing hyperlipidemia, which are of importance for the development and utilization of pomelo peel resources. The languages are suggested to be greatly improved while some descriptions are tediously long. Moreover, there are major points as shown below that could be greatly addressed to further improve the manuscript, which can be accepted in Molecules after major revision.

Major points:

1.     The abstract should compactly and clearly summarize the main ideas of the paper. Thus, I suggest the authors improve the abstract greatly referring to these articles that have been published in Molecules.   

2.     The part of “3.1. Qualitative analysis of naringin” is also tediously long since naringin is well-known flavonoid glycoside natural product. A brief characterization of naringin by spectral analysis of NMR, UV, IR, and HR-MS data is completely enough. Meanwhile, the UV, IR, NMR, and HR-MS spectra are suggested to be assigned in the supporting information. By the way, the 13C NMR data of naringin are suggested to be collected.

3.     The part of “3.2. Quantitative analysis of naringin” is suggested to be combined with that of “Qualitative analysis of naringin”.

4.     Some figures in this paper are not clear and those with high-quality should be provided, including Figures 4, 6, and 7.

5.     Tables 1 and 2 belong to the experimental methods and are also suggested to be assigned in the supporting information.

6.     In Figures 8-10, the illustrations of “**” and “##” should also be provided, which are suggested to be merged. 

Minor points:

1.     Line 4, “a, b” should use superscript.

2.     Line 25, (GSH-Px) Low---(GSH-Px), Low

3.     Line 30, P<0.01--- P (using italic) < 0.01, please check the whole manuscript.

4.     Line 34, kidney and--- kidney, and

5.     Line 117, check this paragraph.

6.     Line 156, OD550--- OD550 (subscript)

7.     Line 192, v/v should be italic. Please check the whole manuscript.

8.     Line 237, cm-1--- cm-1 (superscript).

9.     Line 248, c=c---C=C

10.  Line 256, DMSO-d6, 7.40-7.42 (dd, J =2.72, 8.3 Hz, 2H). Please check the whole paragraph.

11.  Line 293, m/z should be italic. [M + H]+, [M + Na]+, [M + K]+

12.  Line 329, 4 wks---4 weeks

13.  Line 499, medium- and high--- medium-, and high

14.  Line 559, please check this paragraph.

15.  Line 625, the nomenclatures are not full, such as PPAR, FAS, etc. Please check the whole manuscript.

16.  please check the styles of these references according the journal requirements.

Author Response

Dear Reviewer and Editors,

Thank you for giving us an opportunity to revise the manuscript (Molecules-205039)! We are sincerely grateful for these critical comments and thoughtful suggestions on our manuscript. These comments are all valuable and very helpful for revising and improving our article. We have studied these comments and suggestions carefully and have made careful correction. Revised portion are marked in the article. In addition, we have asked an English language editing service to improve the writing in every part of manuscript. The main corrections in the paper and the responses to the comments are as following:

Reviewer #3:

Comments

The paper entitled “Extraction of naringin and evaluation of its hypolipidemic activity” written by Xiaolei Yu and co-workers purified and characterized naringin from the pomelo peel and evaluated its hypolipidemic properties and antioxidant capacity. These findings will help clarify the efficacy and mechanism of naringin in preventing hyperlipidemia, which are of importance for the development and utilization of pomelo peel resources. The languages are suggested to be greatly improved while some descriptions are tediously long. Moreover, there are major points as shown below that could be greatly addressed to further improve the manuscript, which can be accepted in Molecules after major revision.

Major points:

  1. The abstract should compactly and clearly summarize the main ideas of the paper. Thus, I suggest the authors improve the abstract greatly referring to these articles that have been published in Molecules.

Response: Thank you for your professional suggestions. According to reviewer’s suggestion, we have made the modification in line 17-40 Page 1 of the revised manuscript.

  1. The part of “3.1. Qualitative analysis of naringin” is also tediously long since naringin is well-known flavonoid glycoside natural product. A brief characterization of naringin by spectral analysis of NMR, UV, IR, and HR-MS data is completely enough. Meanwhile, the UV, IR, NMR, and HR-MS spectra are suggested to be assigned in the supporting information. By the way, the 13C NMR data of naringin are suggested to be collected.

Response: Thank you for your professional suggestions. According to reviewer’s suggestion, we have made the modification in line 251-269 Page 6 and Supporting Information of the revised manuscript.

  1. The part of “3.2. Quantitative analysis of naringin” is suggested to be combined with that of “Qualitative analysis of naringin”.

Response: Thank you for your professional suggestions. According to reviewer’s suggestion, we have made the modification in line 251-269 Page 6 of the revised manuscript.

  1. Some figures in this paper are not clear and those with high-quality should be provided, including Figures 4, 6, and 7.

Response: Thank you for your professional suggestions. According to reviewer’s suggestion, we have made the modification in line424Page 11and in line 430 Page 12 and Supporting Information of the revised manuscript.

  1. Tables 1 and 2 belong to the experimental methods and are also suggested to be assigned in the supporting information.

Response: Thank you for your professional suggestions. According to reviewer’s suggestion, we have assigned Tables 1 and 2 in the supporting information.

  1. In Figures 8-10, the illustrations of “**” and “##” should also be provided, which are suggested to be merged. 

Response: Thank you for your professional suggestions. According to reviewer’s suggestion, we have made the modification in line 479 Page 14 of the revised manuscript.

Minor points:

  1. Line 4, “a, b” should use superscript.

Response: Thank you for your professional suggestions. According to reviewer’s suggestion, we have made the modification in line 4 Page 1 of the revised manuscript.

  1. Line 25, (GSH-Px) Low---(GSH-Px), Low

Response: Thank you for your professional suggestions. According to reviewer’s suggestion, we have made the modification in line 25 Page 1 of the revised manuscript.

  1. Line 30, P<0.01--- P(using italic) < 0.01, please check the whole manuscript.

Response: Thank you for your professional suggestions. According to reviewer’s suggestion, we have made the modification in line 30 Page 1 and check the whole manuscript of the revised manuscript.

  1. Line 34, kidney and--- kidney, and

Response: Thank you for your professional suggestions. According to reviewer’s suggestion, we have made the modification in line33- 34 Page 1 of the revised manuscript.

  1. Line 117, check this paragraph.

Response: Thank you for your professional suggestions. According to reviewer’s suggestion, we have check this paragraph in line 126-134 Page 3 of the revised manuscript.

  1. Line 156, OD550--- OD550(subscript)

Response: Thank you for your professional suggestions. According to reviewer’s suggestion, we have made the modification in line 171 Page 4 of the revised manuscript.

  1. Line 192, v/v should be italic. Please check the whole manuscript.

Response: Thank you for your professional suggestions. According to reviewer’s suggestion, we have made the modification in line 189 Page 4 and check the whole manuscript of the revised manuscript.

  1. Line 237, cm-1--- cm-1 (superscript).

Response: Thank you for your professional suggestions. According to reviewer’s suggestion, we have made the modification in the supporting information.

  1. Line 248, c=c---C=C

Response: Thank you for your professional suggestions. According to reviewer’s suggestion, we have made the modification in the supporting information.

  1. Line 256, DMSO-d6, 7.40-7.42 (dd, J=2.72, 8.3 Hz, 2H). Please check the whole paragraph.

Response: Thank you for your professional suggestions. According to reviewer’s suggestion, we have made the modification in the supporting information.

  1. Line 293, m/z should be italic. [M + H]+, [M + Na]+, [M + K]+

Response: Thank you for your professional suggestions. According to reviewer’s suggestion, we have made the modification in the supporting information.

  1. Line 329, 4 wks---4 weeks

Response: Thank you for your professional suggestions. According to reviewer’s suggestion, we have made the modification in line 279 Page 7 of the revised manuscript.

  1. Line 499, medium- and high--- medium-, and high

Response: Thank you for your professional suggestions. According to reviewer’s suggestion, we have made the modification in line 440 Page 13 of the revised manuscript.

  1. Line 559, please check this paragraph.

Response: Thank you for your professional suggestions. According to reviewer’s suggestion, we have made the modification in line 489-504 and check this paragraph of the revised manuscript.

  1. Line 625, the nomenclatures are not full, such as PPAR, FAS, etc. Please check the whole manuscript.

Response: Thank you for your professional suggestions. According to reviewer’s suggestion, we have made the modification in line 559 Page 16 and check this paragraph of the revised manuscript.

  1. please check the styles of these references according the journal requirements.

Response: Thank you for your professional suggestions. According to reviewer’s suggestion, we have check the styles of these references according the journal requirements of the revised manuscript.

We have tried our best to improve the manuscript and made some changes in the manuscript. At last, we would like to express our great appreciation to you for comments on our paper. We feel so sorry that so much of your previous time was wasted on our paper revision. We appreciated for Editors and Reviewers’ warm work earnestly, and hope that the correction will meet with approval.

Once again, thank you very much for your comments and suggestions.

With kind regards,

Xiaolei Yu

Correspondence: Lei Zhang: [email protected]. MOE Key Laboratory for Nonequilibrium Synthesis and Modulation of Condensed Matter, School of Physics, Xi'an Jiaotong University, No. 28, Xianning West Road, Xi’an, Shaanxi Province 710049, People’s Republic of China. Tel: +86-029-82668634

Reviewer 4 Report

The manuscript provided some valuable data on Extraction of naringin and its hypolipidemic activity. However, some issues should be revised.

1.     The notes of Figure1-5, and Figure 7-10 need to be strengthened for their self-evident nature. And the notes of table 3-4 need to be revised for the accuracy, ie: In table 3, “Note:n = 5 compared with blank control. (*P < 0.05, **P < 0.01).” should be “**P < 0.01, compared with blank control.”…

2.     The Figure 6A should be shown as line chart than histogram

3.     The description of some results such as “FAS mRNA expression” …need to be corrected.

4.     “Conclusion” should be more accurate based on the results of present experiment.

5.     In 2.4, “(Certificate No. SCXK (Liao) 2019-003)” might be “[Certificate No. SCXK (Liao) 2019-003]”, and In 2.10, “means ± standard error (±SD)” should be “means ± standard error (±SE)”,

Author Response

Dear Reviewer and Editors,

Thank you for giving us an opportunity to revise the manuscript (Molecules-205039)! We are sincerely grateful for these critical comments and thoughtful suggestions on our manuscript. These comments are all valuable and very helpful for revising and improving our article. We have studied these comments and suggestions carefully and have made careful correction. Revised portion are marked in the article. In addition, we have asked an English language editing service to improve the writing in every part of manuscript. The main corrections in the paper and the responses to the comments are as following:

Reviewer #4:

Comments and Suggestions for Authors

The manuscript provided some valuable data on Extraction of naringin and its hypolipidemic activity. However, some issues should be revised.

  1. The notes of Figure1-5, and Figure 7-10 need to be strengthened for their self-evident nature. And the notes of table 3-4 need to be revised for the accuracy, ie: In table 3, “Note:n = 5 compared with blank control. (*P < 0.05, **P < 0.01).” should be “**P < 0.01, compared with blank control.”

Response: Thank you for your professional suggestions. According to reviewer’s suggestion, we have made the modification in line 275-276 Page 7,line 287 Page7,line 374 Page9, line 432-435 Page 12,line 481-486 Page 14 and Supporting Information.

  1. The Figure 6A should be shown as line chart than histogram

Response: Thank you for your professional suggestions. According to reviewer’s suggestion, we have made the modification in line 423 Page 11 of the revised manuscript.

  1. The description of some results such as “FAS mRNA expression” …need to be corrected.

Response: Thank you for your professional suggestions. According to reviewer’s suggestion, we have made the modification in line 438-444 Page 13 of the revised manuscript.

  1. “Conclusion” should be more accurate based on the results of present experiment.

Response: Thank you for your professional suggestions. According to reviewer’s suggestion, we have made the modification in line 532-543 Page 15 and line 544-546 Page 16 of the revised manuscript.

  1. In 2.4, “(Certificate No. SCXK (Liao) 2019-003)” might be “[Certificate No. SCXK (Liao) 2019-003]”, and In 2.10, “means ± standard error (±SD)” should be “means ± standard error (±SE)”,

Response: Thank you for your professional suggestions. According to reviewer’s suggestion, we have made the modification in line 127-129 Page 3 and line 246-247 Page 5 of the revised manuscript.

Male SPF Kunming (KM) mice (each weighing 25 ± 5 g), procured from the Exper-imental Animal Center of Jinzhou Medical University (Laboratory Animal Production License Certificate No. SCXK (Liao) 2019-0003)and(Laboratory animal use permit Certificate No. SYXK (Liao) 2019-0007),were used in this study.

We have tried our best to improve the manuscript and made some changes in the manuscript. At last, we would like to express our great appreciation to you for comments on our paper. We feel so sorry that so much of your previous time was wasted on our paper revision. We appreciated for Editors and Reviewers’ warm work earnestly, and hope that the correction will meet with approval.

Once again, thank you very much for your comments and suggestions.

With kind regards,

Xiaolei Yu

Correspondence: Lei Zhang: [email protected]. MOE Key Laboratory for Nonequilibrium Synthesis and Modulation of Condensed Matter, School of Physics, Xi'an Jiaotong University, No. 28, Xianning West Road, Xi’an, Shaanxi Province 710049, People’s Republic of China. Tel: +86-029-82668634

Round 2

Reviewer 1 Report

The manuscript Molecules-205039, revised by the researcher's is satisfactory and acceptable in its present form because they had made changes in the manuscript and make it more understandable for reader's. 

Thanks for 

Reviewer 2 Report

All the comments have been addressed by the authors. I have no further comments.

Reviewer 3 Report

The revised paper have been improved well. Thus, I think it could be accepted after language checking. Thank you.